# Multiscale Entropy of Cardiac and Postural Control Reflects a Flexible Adaptation to a Cognitive Task

**DOI:** 10.3390/e21101024

**Published:** 2019-10-21

**Authors:** Estelle Blons, Laurent M. Arsac, Pierre Gilfriche, Veronique Deschodt-Arsac

**Affiliations:** 1Univ. Bordeaux, CNRS, Laboratoire IMS, UMR 5218, 33400 Talence, France; laurent.arsac@u-bordeaux.fr (L.M.A.); pierre.gilfriche@u-bordeaux.fr (P.G.); veronique.arsac@u-bordeaux.fr (V.D.-A.); 2CATIE—Centre Aquitain des Technologies de l’Information et Electroniques, 33400 Talence, France

**Keywords:** heart rate variability, posture, entropy, complexity, cognitive task

## Abstract

In humans, physiological systems involved in maintaining stable conditions for health and well-being are complex, encompassing multiple interactions within and between system components. This complexity is mirrored in the temporal structure of the variability of output signals. Entropy has been recognized as a good marker of systems complexity, notably when calculated from heart rate and postural dynamics. A degraded entropy is generally associated with frailty, aging, impairments or diseases. In contrast, high entropy has been associated with the elevated capacity to adjust to an ever-changing environment, but the link is unknown between entropy and the capacity to cope with cognitive tasks in a healthy young to middle-aged population. Here, we addressed classic markers (time and frequency domains) and refined composite multiscale entropy (MSE) markers (after pre-processing) of heart rate and postural sway time series in 34 participants during quiet versus cognitive task conditions. Recordings lasted 10 min for heart rate and 51.2 s for upright standing, providing time series lengths of 500–600 and 2048 samples, respectively. The main finding was that entropy increased during cognitive tasks. This highlights the possible links between our entropy measures and the systems complexity that probably facilitates a control remodeling and a flexible adaptability in our healthy participants. We conclude that entropy is a reliable marker of neurophysiological complexity and adaptability in autonomic and somatic systems.

## 1. Introduction

Physiological control is critical for health and well-being in humans, as it contributes to maintaining homeostasis and the adoption of adequate behaviors. Effective control takes place across intricate networks spanning many neural structures and operating across many time scales. These networks are dynamically organized to respond to internal and external stimuli. The coordinate functioning of the many constitutive components, their multiple interactions within and between systems, and the presence of overlapping control loops have promoted the conceptualization of nonlinear systems, exhibiting complexity [1]. 

The emergent field of systems physiology exploits the idea that complexity is mirrored in the temporal structure of a system’s output variable. By analyzing physiological time series generated by control systems (e.g., the autonomic control of heart rate [1,2] or the somatic control of postural sway when standing upright [3,4]), researchers have discovered a preserved richness of the information carried by the output signals across multiple timescales. This richness in physiological signals can be assessed based on sample entropy [5], a measure of the irregularity of a time series obtained by calculating the probability that segments (also called vectors) of similar m samples remain similar when the segment length increases to m+1. Entropy-based complexity metrics relate to the information content of a signal by quantifying the degree of regularity or predictability over one or more scales of time. To address this issue, Costa et al. [1,2] have proposed a multiscale entropy (MSE) method that consists of a coarse-graining process and sample entropy computations to measure the complexity of a time series at different temporal scales.

The true strength of this method lies in considering the sample entropy value over multiple time scales rather than one unique scale. By considering many scales, one can evaluate how far a system deviates both from emitting white noise (meaning a degraded network organization) and emitting a very regular signal, which is interpreted as too strict an organization and a lack of flexibility. 

In agreement with these interpretations, experimental applications have demonstrated a degraded entropy in cardiac and postural dynamics associated with frailty, aging, impairments, or diseases [1,3,4,6,7,8,9,10,11,12,13,14,15,16,17,18,19]. By contrast, high entropy is generally associated with an elevated capacity to adjust to an ever-changing environment [8], and elevated values are often observed in young healthy people [1]. 

During a dual-task protocol, the degradation of entropy in postural sway is exacerbated in aged people [3,8], thus indicating a failure in the dynamic re-organization of control. A similar phenomenon was observed in cardiovascular control when comparing nocturnal and diurnal MSEs of heart rate dynamics [1]. During waking periods, complexity raised in young individuals but vanished in old-age individuals, which lets the authors suppose that environmental stimuli (and the need for multi-tasking) may exceed a system’s capacity, thus prohibiting an adequate re-organization in aged people.

One can ask whether stimuli not exceeding a system’s capacity leads to an adequate re-organization of physiological control, and whether this is reflected in a greater signal entropy. In other words, it is unclear to date if the capacity to cope with a cognitive task in a healthy young to middle-age population is reflected in the entropy of a control system’s output, while a degraded entropy seems to be the rule among old-aged individuals.

The aim of the present study is to assess the dynamic organization of control when performing cognitive tasks using the temporal behavior of heart rate and postural dynamics according to a multiscale entropy approach. We hypothesized that entropy would increase during the cognitive tasks, thus highlighting a flexible adaptation of neurophysiological control in our healthy participants. 

## 2. Materials and Methods

### 2.1. Population

Thirty-four volunteers (8 women, 26 men) gave their written informed consent to participate in the present study in accordance with a local institutional review board policy and with the principles of the Declaration of Helsinki. The mean and standard deviation values of participants’ age and body mass indexes were 30.5 ± 14.0 years (range: 18–59) and 21.1 ± 1.9 kg/m^2^, respectively. Among the women, four were using oral contraceptives, five reported being in the follicular phase of their menstrual cycle and three were in the luteal phase. All volunteers had a university education.

None of the participants reported neurological or physiological disorders. Participants were asked to avoid alcohol and caffeinated beverages for the 12 h preceding the experiment, but also to abstain from heavy physical activity.

### 2.2. Protocol

The experimental protocol included recordings of heart rate dynamics and postural dynamics, according to reference (Ref) and cognitive tasks (Cog). Recordings of heart rate dynamics lasted 10 min during which the participants were sitting down in a quiet environment, breathing normally (at a spontaneous rate), and either facing a blank computer screen (Ref), or performing cognitive tasks displayed on the screen (Cog). Recordings of postural dynamics lasted 51.2 s, during which the participants had to stand upright on a force platform, either looking at a black cross 4 m ahead (Ref), or performing a cognitive task displayed on a screen 4 m ahead (Cog). This study followed a randomized crossover design in which participants executed either cardiac or postural measurements first, and, in each of these two blocks of measurements, either Ref or Cog was executed first.

### 2.3. Recordings of RR Interval Time Series

Cardiac interbeat (RR interval) time series were recorded from a bipolar electrode transmitter belt Polar H7 (Polar, Finland) fitted to the chest of the subject and connected to an iPod (Apple, Cupertino CA, USA) via Bluetooth. A smartphone application was used to continuously store the transmitted RR intervals. About 500–600 successive RR intervals were recorded over 10 min, the exact length of the RR interval time series depending on the average heart rate of each participant. For further analyses, the RR interval time series were exported to Matlab (Matworks, Natick, MA, USA).

### 2.4. Recordings of Center of Pressure Time Series

Anteroposterior (AP) and mediolateral (ML) postural sway was assessed from the center of pressure (COP) trajectory and recorded by a platform equipped with three strain gauges (Winposturo, Medicapteurs, 40 Hz/16b, Balma, France). Participants stood barefoot with feet abducted at 15° from the median line and heels separated by 4 cm. Participants’ eyes were open and their arms hung loosely at their sides. COP trajectories were recorded at a sampling frequency of 40 Hz for 51.2 s (thus providing 2048 data points). The AP and ML time series were exported to Matlab (Matlab R2017b, Mathworks) for further analyses.

### 2.5. Cognitive Tasks

During Cog, participants performed cognitive tasks chosen to solicit frontal cortical lobes, cerebral areas where executive functions operate [20,21]. 

During the entire 10-min recordings of heart rate dynamics, participants performed four tests that followed one another in this order: the Stroop Color and Word Test (SCWT) [22], the Hayling Sentence Completion Test (HSCT) [23], a visual version of the Paced Auditory Serial Addition Test (PASAT) [24], and a semantic fluency task [25]. In order to ensure that participants remained silent during these tests, they wrote their answers to the test. The durations of each task were the following: 3 min for the SCWT, 2.5 min for the HSCT, 3 min for the PASAT, and 1.5 min for the semantic fluency task. SCWT is a task that forces inhibition of cognitive interference, which occurs when the processing of a stimulus feature affects the simultaneous processing of another attribute of the same stimulus [22]. The HSCT taps into response initiation and response inhibition [23]. The PASAT requires attentional functioning, working memory, and information processing speed [24]. The semantic fluency task consisted of spontaneous narration about a given topic (e.g., supermarkets) [25]. 

Due to the short duration of the recordings of postural dynamics (51.2 s), SCWT alone was administrated. Participants answered verbally.

### 2.6. Analysis of RR Interval Time Series: Classic Indices

Due to technical issues, two participants (one woman and one man) were excluded from the RR interval time series analyses. All computations were performed in Matlab using available functions and custom-designed routines. The raw data of heart rate variability (HRV; RR interval time series) were inspected for artifacts. Occasional ectopic beats (irregularity of the heart rhythm involving extra or skipped heartbeats such as extrasystole and consecutive compensatory pause), were visually identified and manually replaced with interpolated adjacent RR interval values. Classic indices were then calculated in time and frequency domains. The mean of RR interval values was calculated. The root mean square of successive differences (RMSSD) was obtained by calculating the first difference, a discrete analog of the first derivative, which is a standard method for removing slow varying trends in a signal and highlights the power of high-frequencies that are associated with parasympathetic modulations of the heart rate [26]. In the frequency domain, discrete Fourier transform was performed after 4 Hz resampling using a cubic spline interpolation. The computation of signal power in fixed bands between 0.04 and 0.15 Hz for the low frequencies (LFs) and between 0.15 and 0.4 Hz for the high frequencies (HFs), allowed the calculation of the ratio LF/HF (an index of the sympathovagal balance) [26].

### 2.7. Analysis of Center of Pressure Time Series: Classic Indices

To evaluate the main features of postural control, here we computed the 95% confidence ellipse area, which is expected to enclose approximately 95% of the points on the COP path [19]. As well, the average velocity along the AP and the ML axes was computed. In the frequency domain, the spectral energy was assessed on ML and AP axes based on the power spectral density (PSD) obtained with fast Fourier transform.

### 2.8. Analysis of Complexity: Entropy Indices

The refined composite multiscale entropy (RCMSE) [27] was computed from both RR interval time series and postural time series in order to investigate signal complexity. As mentioned by Wu et al., the RCMSE method proposes improve the MSE method for short time series [2,27] by increasing the accuracy of entropy estimation and reducing the probability of inducing undefined entropy [27]. Undefined entropy may result from computations of short time series where no template segments (vectors) are matched to one another.

In brief, in the original MSE algorithm [1,2], the analyzed time series $x=\left\{x_{1},\;x_{2},\;\ldots ,\;x_{N}\right\}$ is coarse grained using non-overlapping windows to obtain the representation of the original time series at different time scales τ. The algorithm detects how many segments (vectors) of size m remain similar at size m+1 in the time series. Hence, the number of matched vector pairs indicates the level of signal regularity. Due to a reduction of the original signal by a factor of τ, the time series at large scale factors is composed of much fewer data points that the original one [27,28]. This is a concern for the accuracy of entropy calculation, mainly in short time series. A first attempt to address this accuracy concern was the development of composite multiscale entropy (CMSE) [29], whose main gain relies on considering all possible starting points at a given scale for the coarse-grained process, then calculating the averaged sample entropy for each scale. It was observed that CMSE, despite possessing a greater accuracy, increases the probability of inducing undefined entropy. To address this particular concern, Wu et al. (2014) [27] developed refined composite multiscale entropy (RCMSE), a method that uses the number of matched vector pairs for each scale factor τ and also for all $\left(k\right)\;\tau$ coarse-grained time series. Hence, it is unlikely even for short time series that the sum of matched vector pairs are zeros.

Briefly, the RCMSE algorithm consists of the following procedures (see detailed method in [27]):
At each scale factor of τ, the number of matched vector pairs $n_{k,\tau }^{m+1}$ and $n_{k,\tau \;}^{m}$ is calculated for all $\left(k\right)\;\tau$ coarse-grained series, with m corresponding to the sequence length considered. In the present study, m=2. The RCMSE at a scale factor of τ is provided as follows, with r corresponding to the tolerance for matches. In the present study, r=0.15 of the standard deviation of the initial time series x [30].
(1)$$RCMSE\left(x,\;\tau ,\;m,\;r\right)=-ln\left(\frac{\sum_{k=1}^{\tau }n_{k,\tau }^{m+1}}{\sum_{k=1}^{\tau }n_{k,\tau }^{m}}\right)$$

The length of the original time series determines the largest analyzed scale [1,27,31]. In this study, RCMSE was assessed over a range of scales from 1 to 4 for RR interval time series and over a range of scales from 1 to 14 for postural times series, a difference that was due to different sample sizes of RR interval (500 to 600 samples) and postural (2048 samples) times series. 

The RCMSE curve is obtained by plotting entropy values for each coarse-grained time series as a function of scales. The cardiac entropy index (E_C_) and postural entropy index (E_P_) are the area under the corresponding RCMSE curves (areas calculated using the trapezoidal rule) (Figure 1) [1,27]. As recommended by Gow et al. [31], entropy indices were computed after pre-processing time series using empirical mode decomposition (EMD) [32]. EMD decomposes a signal into a sum of intrinsic mode functions (IMFs) and a residual trend. This residual trend was subtracted to remove the drift, which has been identified as a source of error in entropy assessments [31].

We tested the hypothesis that the complexity of our time series is encoded in the sequential ordering, and that this ordering is not fortuitous. For that, we built surrogate time series by shuffling the sequence of data points (randomly reordering). RCMSE curves are presented comparatively (see the figure in Section 3.2).

### 2.9. Statistical Analyses

All statistical procedures were conducted by use of XLSTAT (Addinsoft, 2019, XLSTAT statistical and data analysis solution, Long Island, NY, USA). Classic and entropy indices were tested for normality (Shapiro-Wilk test). These indices were compared between Ref and Cog conditions (two-tail *t*-test or Wilcoxon test). Following the American Statistical Association statement on statistical significance and *p*-values, we did not base our scientific conclusions only on whether a *p*-value passes a specific threshold (usually, *p* < 0.05). Measures of detection sensitivity theory were additionally employed to assess sensitivity and specificity of the obtained indices, including the receiver operating characteristic (ROC) [33]. The area under the ROC curve indicates the probability that the index will assign a higher value to a positive instance than to a negative one [34]. Youden’s index $\left(J=Sensitivity+Specificity\;-1\right)$ assesses the performance of the detector. 

## 3. Results

Figure 2 shows typical signal outputs from the two explored neurophysiological systems obtained for a single participant: RR interval times series under reference (Ref) and cognitive (Cog) conditions are shown in the top panel; anteroposterior (AP) and mediolateral (ML) time series of the COP trajectory are reported below in middle and bottom panels respectively.

Mean values of classic and entropy indices derived from the signals obtained from our participants are reported in Table 1.

### 3.1. Classic Indices in Temporal and Frequency Domains

The mean RR decreased (heart rate increased) under the Cog conditions (*p* < 0.001, two-tail Wilcoxon test). 

None of the classic temporal (RMSSD) or frequency-derived heart rate variability (HRV) indices (LF, HF, LF/HF) differed between Ref and Cog, meaning that power at any given frequency did not change during Cog. Regarding posture, no difference in 95% confidence ellipse or total PSD-derived energy was observed in the COP displacement signals, while the COP velocity differed (AP *p* < 0.001, two-tail Wilcoxon test and ML *p* = 0.046, two-tail Wilcoxon test). 

### 3.2. Entropy Indices

As expected, the RCMSE curves for the shuffled (randomly ordered) time series markedly differed from the RCMSE curves for the original time series (Figure 3). Entropy as a function of scales exhibited a monotonic decrease in shuffled time series, which is characteristic of random (white) noise [1,3]. By contrast, heart rate and postural dynamics exhibited typical behavior of a complex system, where the richness of carried information (as represented by entropy at a given scale) do not vanish when observed in longer timescales. 

The main entropy index values (E_C_ and E_P_) are presented in Table 1. As a main finding here, the E_C_ index obtained during Cog was higher than the index obtained during Ref (*p* = 0.016, two-tail Wilcoxon test).

As well, along the AP axis where most of the postural (dys)regulation occurs [35,36], the E_P_ index obtained during Cog was higher than the index obtained during Ref (*p* < 0.001, two-tail t-test). The ML E_P_ indices did not differ between Ref and Cog (Table 1).

### 3.3. ROC Curves Analysis

The ROC curves are shown in Figure 4, and the corresponding areas under the curves (AUC) and the Youden’s indexes are reported in Table 2. The greatest AUC was obtained for entropy of both cardiac (0.67) and postural (0.72) time series, thus indicating that entropy showed a higher probability to assign a higher value to a positive instance than to a negative one.

## 4. Discussion

In this study we attempted to highlight the possible links between entropy measurements in two distinct neurophysiological networks and the systems complexity that probably facilitates the auto-organization and flexible adaptability in our healthy participants.

The main finding was that performing cognitive tasks resulted in a heightened entropy in heart rate and postural oscillations in young healthy people when compared to quiet conditions, as hypothesized. This may demonstrate that eliciting brain activity induced a remodeling in involuntary control networks, leading to a greater richness in signal information. This result is coherent with a great flexibility in our healthy young participants, which contrasts with a decline in entropy reported in older-aged individuals during a dual-task [3,8]. Both the elevation of entropy during cognitive tasks and the fact that two different neurophysiological systems behave in the same way represent original findings in the present study.

The link between central (cognitive) and peripheral regulations has been widely acknowledged. As a topic of growing interest, heart–brain interactions rely on a complex network of interconnected neural structures in the central autonomic network, whose functions are organized at the forebrain, brainstem, and spinal levels [37,38,39,40,41]. As shown by functional imaging, cortical and subcortical brain activities influence autonomic outflow to the periphery [42,43,44,45]. In our conditions, executive functions and associated prefrontal regions were involved during the imposed cognitive tasks. It is likely that the recruitment of brain regions reverberated throughout the autonomic outflow, as reflected in the heightened complexity revealed here by the RCMSE metrics in heart rate dynamics.

The rise in cardiac entropy is a marker of complex dynamics, which has been shown to reflect an underlying highly dimensional system with multiple interacting components associated with a high level of functionality [46,47]. Therefore, we can suggest that the observed increase in entropy during the cognitive tasks relies on remodeling and adaptability from the baseline, triggered by the recruitment and the interactions between brain components. This capacity to reorganize the control network in such a way that complexity is increased underscores a system’s reserve that is not exhausted by any of our conditions [1]. This observation is in agreement with Costa et al. [1], who demonstrated that cardiac entropy (MSE) rose in healthy young people when facing diurnal challenges (waking period) that are absent during the night (sleep period). Cardiac entropy failed to increase comparatively in older-aged subjects. Other complexity metrics of HRV dynamics, such as fractal long-range properties in the temporal structure, provided additional evidence that cardiac complexity rises when the brain performs executive functions, which was reflected in clearer 1/f noise [48]. Yet, entropy metrics may provide greater reliability for analyzing complexity from short-term HRV, because fractal properties are mainly dictated by power versus frequency characteristics of two dominant oscillators relying on vagal and sympathetic controls [49]. Hence, the “true fractal” component of the spectrum should be assessed only on frequencies < 0.04 Hz, which requires long-lasting RR interval time series recordings [50]. Noticeably, RCMSE provided satisfactory results for the presence of a complex (1/f) system’s behavior in our conditions (10 min recordings).

MSE has traditionally been computed to study the COP trajectory as an index of complexity in the neurophysiological control of posture, and a number of recommendations have been very useful in this domain [31]. The pre-processing of the COP signal in the present study (EMD filtering) is part of the cautious approach that is recommended. While it is usually reported that dual-tasking provides a decline in entropy among older-age individuals, we clearly show in this study that COP entropy rose (rather than dropped) in our young healthy participants. This highlights an adaptive capacity when recruiting cognitive functions and their related brain regions, which contrasts with the degraded [3,8], but reversible [51], flexibility in older-aged dual-tasking. 

It is not trivial to observe a similar behavior (the increase of entropy) in the present study both in relation to cardiovascular and postural control among our participants as a response to the cognitive task. These systems are markedly different; while the cardiac control relies on neurovisceral integration, the postural sway results from the somatosensory integration of exteroceptive and proprioceptive information. The rise in entropy therefore seems ubiquitous, and as such may reflect an adequate dynamic organization of neurophysiological control with improved interactions both within and between systems, whatever their neural structures.

Although the discovery of an increase in systems complexity in response to cognitive tasks is original in the present study, previous recent experiments have demonstrated that specific interventions may improve a degraded complexity. In humans, the capacity to restore a degraded postural complexity in aged people has been shown following mind–body interventions [4,7,9]. As well, walking arm-in-arm has recently been shown as an efficient way to restore walking complexity among older-aged individuals [52]. For years, degraded complexity markers (fractal or entropy metrics) in physiological signal outputs have been associated with impaired physiological control. The present study participates in the recent demonstrations of a heightened complexity marker indicating improved neurophysiological control. 

## 5. Conclusions

By comparing quiet and cognitive task conditions, MSE-based metrics emphasize an adaptive systems capacity and a potential remodelling of cardiac and postural control systems under temporary states of cognitive tasks. The rise in entropy associated with cognitive functions, which contrasts with a decline reported in old people, illustrates improved interactions across brain regions and peripheral control loops, leading to a great richness in regulatory information. This demonstrates that the functional reserve capacity was not reached by our young healthy participants under our conditions. The issue of overwhelmed control systems in healthy young people confronted with cognitive tasks remained to be explored, through varying cognitive workloads or combining them with challenging emotions (e.g., stress), for example. It would be great to observe that whether, after heightening entropy in young people, more strenuous cognitive loads (with or without additional stressors) could push control systems to their adaptive limits, and whether this is reflected by a decline in entropy. It is unknown if the two distinct neurophysiological systems will keep demonstrating a similar behaviour when one faces such gradual challenges. With further study, even more credit could be gained towards entropy metrics and their capacity to faithfully reflect tight adjustments in complex physiological systems during gradual stimulations.

## 6. Limitations

Despite appealing results, the present study was not without limitations. The number of participants might have been augmented, in particular the number of females offering the opportunity to explore sexual dimorphism, as noted elsewhere [53]. Regarding gender, it was noted that even a methodological choice for MSE may influence physiological interpretations due to sex-related differences in cardiovascular dynamics [30]. While we used a fixed tolerance r at all scales in this study, an alternative method suggests adjusting the tolerance to the standard-deviation changes after coarse graining [30]. This might improve MSE estimation of heart rate and could be tested on the present data. It is presently unlikely that adopting an alternative (among many possible) usage of MSE could change the main conclusions of the present study; indeed, RCMSE on shuffle time series was computed here, clearly highlighting the distance from a random neurophysiological control and the capacity of RCMSE to distinguish quiet and cognitive task conditions (Figure 3). Finally, we have no explanation for the lack of change in ML entropy due to the cognitive task during postural regulation. Further studies are needed to explore the potential role of certain instances that could dominantly aggregate AP information, making complex AP regulations more responsive than ML.

## Figures and Tables

**Figure 1 entropy-21-01024-f001:**
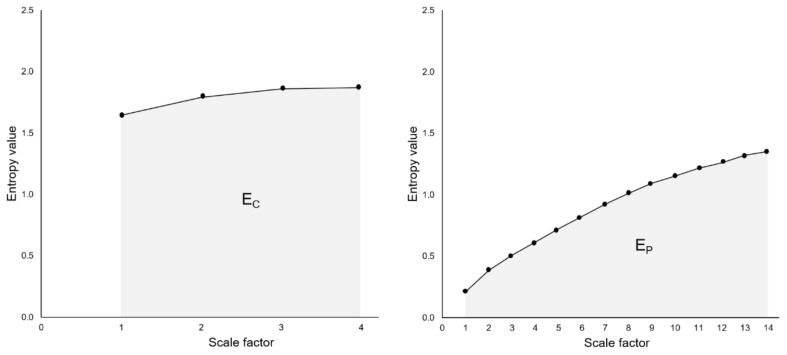
Cardiac entropy index (E_C_, **left**) and postural entropy index (E_P_, **right**), calculated from the areas under the refined composite multiscale entropy (RCMSE) curves.

**Figure 2 entropy-21-01024-f002:**
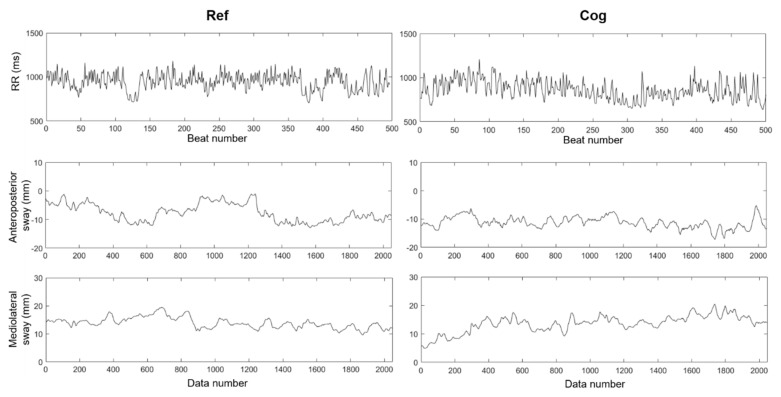
Top: RR interval time series from a representative participant in reference (Ref, **left**) and cognitive (Cog, **right**) conditions. Middle and bottom: anteroposterior (AP, **middle**) and mediolateral (ML, **bottom**) center of pressure (COP) time series, the horizontal axes are the same for these plots.

**Figure 3 entropy-21-01024-f003:**
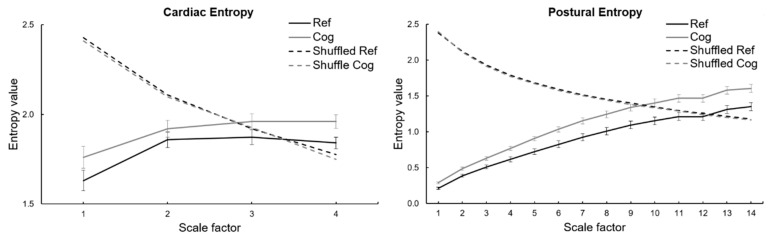
Refined composite multiscale entropy (RCMSE) analysis of RR interval time series (**left**) and center of pressure time series on anteroposterior axis (**right**) during reference (Ref) and cognitive (Cog) conditions. The RCMSE curves were obtained by connecting the group mean values of sample entropy for each scale. The error bars represent standard errors. The RCMSE curves for the surrogate shuffled time series are also presented.

**Figure 4 entropy-21-01024-f004:**
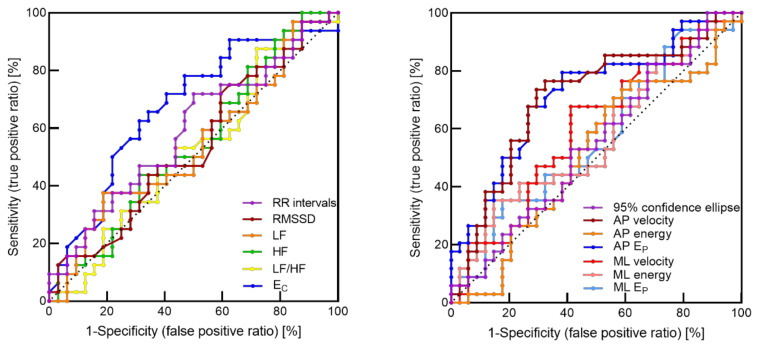
Receiver operating characteristic (ROC) curves (sensitivity vs 1-specificity) for cardiac (**left**) and postural (**right**) indices. RMSSD: root mean square of successive differences; LF: low frequency; HF: high frequency; E_C_: cardiac entropy index; AP: anteroposterior; E_P_: postural entropy index; ML: mediolateral.

**Table 1 entropy-21-01024-t001:** Classic and entropy indices calculated from RR interval time series and from anteroposterior and mediolateral center of pressure time series, during reference and cognitive conditions.

Heart Rate Dynamics	Ref	Cog
RR intervals (ms)	952 ± 120	915 ± 131 ^**^
RMSSD (ms)	58 ± 36	52 ± 30
LFs (ms^2^)	2243 ± 2058	1894 ± 1602
HFs (ms^2^)	1459 ± 1448	1150 ± 1196
LFs/HFs	2.96 ± 3.09	2.82 ± 2.62
E_C_	5.45 ± 0.60	5.75 ± 0.69 ^*^
**Postural Dynamics**	**Ref**	**Cog**
95% confidence ellipse (mm^2^)	217.5 ± 148.5	184.7 ± 103.5
AP velocity (mm·s^−1^)	4.4 ± 1.1	5.1 ± 1.2 ^***^
AP energy (mm^2^)	10.29 ± 19.1	9.04 ± 5.5
AP E_P_	11.81 ± 3.07	14.45 ± 3.27 ^***^
ML velocity (mm·s^−1^)	4.9 ± 1.6	5.3 ± 1.5 ^*^
ML energy (mm^2^)	6.42 ± 3.59	8.00 ± 5.54
ML E_P_	13.99 ± 2.76	14.72 ± 3.03

Values provided are mean ± standard deviation. Ref: reference condition; Cog: cognitive condition; RMSSD: root mean square of successive differences; LFs: low frequencies; HFs: high frequencies; E_C_: cardiac entropy index; AP: anteroposterior; E_P_: postural entropy index; ML: mediolateral. Differences between Ref and Cog are expressed as ^***^
*p* < 0.001, ^**^
*p* < 0.01, ^*^
*p* < 0.05.

**Table 2 entropy-21-01024-t002:** Sensitivity analysis of cardiac and postural indices.

Heart Rate Dynamics	*J*	AUC
RR intervals	0.22	0.59
RMSSD	0.13	0.54
LFs	0.19	0.54
HFs	0.13	0.54
LFs/HFs	0.16	0.52
E_C_	0.31	0.67
**Postural Dynamics**	***J***	**AUC**
95% confidence ellipse (mm^2^)	0.15	0.55
AP velocity	0.44	0.71
AP energy	0.15	0.51
AP E_P_	0.41	0.72
ML velocity	0.27	0.60
ML energy	0.21	0.67
ML E_P_	0.18	0.56

*J*: Youden’s index; AUC: area under the ROC curve; RMSSD: root mean square of successive differences; LFs: low frequencies; HFs: high frequencies; E_C_: cardiac entropy index; AP: anteroposterior; E_P_: postural entropy index; ML: mediolateral.

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
