# Peer review of "Multiscale Entropy of Cardiac and Postural Control Reflects a Flexible Adaptation to a Cognitive Task"

_entropy, 2019, doi:10.3390/e21101024_

Round 1
Reviewer 1 Report
This paper studies cardiac and postural control under quiet and cognitive-loaded tasks in a group of healthy adults employing Entropy. The authors contrast the results with previous foundlings reported in the litterature, also in terms of physiological signal Entropy, on elderly people The authors employ a multiscale Entropy approach claiming significant differences between the two conditions can only be observed with such metrics.
The results enrich the litterature on understanding physiological systems under cognitive-loaded conditions and particularly between young adults and elder people. However, the results are not fully discussed and referenced to the setup methodology. The entropy indexes that play a central role in the paper need also further comparison. Explicitly, the paper should address:
1. Authors should explain why in the postural analysis from AP and ML indexes only AP Entropy is significantly different between the two conditions. The authors mention ‘It is not trivial to observe in the present study a similar behavior (increase in entropy) both in cardiovascular and postural control in our participants in response to the cognitive task. First, this observation strengthens the reliability of entropy markers in output signals because it is unlikely that the observed augmentation in not one, but two systems is fortuitous.’ It is however never discussed why only AP indexes are significantly different while it would be expected that both AP and ML would change significantly.
2. More insight and investigation is required regarding the cognitive tasks employed:
2.1 What would happen if SCWT was exchanged by any other cognitive Task
2.2 What happens if the order of Cognitive tasks is exchanged
2.3 Is it possible to give results related to each of the cognitive tasks applied?
3. A central point in the manuscript is to show that Multiscale entropy is an appropriate measure to put in evidence significant changes in physiological systems otherwise not visible with other indexes. Adding some other entropy-based measures and showing how RCMSE puts in evidence the findings of the project would strengthen the claims of the paper. RCMSE could be better explained particularly by defining ‘undefined entropy’, ‘matched vector pairs’.
Some more detailed comments are given below:
line 18: It would be good to give the whole index name: Refined Composite Multiscale Entropy
line 86: What are the implications of women menstrual state. Could this be discussed or explained later in the paper?
line 111: The length seems to be fixed for COP trajectories (2018 data points). Are RR intervals brought to a final length for final analysis?
line 120: Probably, it would be better to say that COP trajectories are exported to Matlab
line 136: Among the administrated cognitive tests, can it be discussed why this one was chosen.
If the answer given was verbal, was it recorded and transcribed so that the complexity of the cognitive work done can be assessed if possible.#
line 154: Change ‘We computed the 95% confidence ellipse of the area produced by the COP displacements’ to ‘We computed the ellipse fitting, with 95% confidence, the area produced by the COP displacements. ’
line 162: It would be useful to explain/remind what it is meant by undefined entropy
line 173: For self containment, It would be good if the authors have a short explanation of what matched vector pairs are.
line 184: it would be good to give a justification on the contrasting difference on number of scales employed between the two signals. Is it because of the difference in number of samples; is it because of the difference of variability of the signal?
Line 193: “We tested the hypothesis that the complexity of our time series is encoded in the sequential ordering”: Could be possible to analyse the presentation order of cognitive tasks? If it can be deduced how the presentation order affects the results of the study (the result of the Entropy index and what this implies at neurophysiological level).
Line 204: Figure2. Probably a study should be made to analyse what is the number of scales that should be used.
Line 239: Table1. Why LF, HF have ms^2 units? Frequency units are Hertz.
The document reports an ellipse area; where are the area units for the ellipse?
Line 239: Table1. Why are there only significant differences on the Anterioposterior Entropy? Can this be discussed on the paper?
Line 277: change ‘quite’ → ‘quiet’
Line 280: “which contrasts with decline in dual-tasking entropy reported in old people”. Mention the reference, even if it has been given before.
Line 310: “It is worth noting that among all HRV markers computed, only complexity metrics captured the subtle changes in response to the cognitive task” The RR intervals by themselves are significantly different. Please adjust phrasing accordingly.
Line 321: “ dual-tasking elderly” change to “Elderly dual-tasking”
Line 361: “physiological interpretations due to sex-related differences in cardiovascular dynamics”. Can this be discussed also in the main manuscript.
Reviewer 2 Report
This work is about the role of entropy markers in the analysis of physiological signals. In particular, the entropy of heart rate and postural sway time-series recordings is used for detecting and characterizing certain cognitive task conditions. These results are compared with classical time-series descriptors, both in the time and frequency domain. The cognitive tasks considered here follow standard experimental protocols, the same is true for the recording setup. In total 34 subjects were tested, and the role of different markers for discriminating cognitive task conditions from resting-state conditions was investigated. The entropy of the longitudinal signals was quantified using the RCMSE model, and standard implementations of usual time-frequency markers have been used. The "significance" of individual marker was assessed by the p-value of a standard two-tailed t-test. Markers with associated p-values < 0.05 are considered "significant". Statements like this appear frequently in the literature, but nevertheless they are simply misleading, see (Wasserstein & Lazar: The ASA Statement on p-Values: Context, Process, and Purpose, 2016) for a discussion of this topic. Moreover, it appears to me that the authors do use any multiple-testing adjustment of the p-values. No quantitative information about the discriminative power of the markers for separating the two types of conditions is provided.
The main result in the study is that there are indeed some entropy markers that lead to small p-values in this setting, while others do not. A very similar result is obtained for the classical markers. No analysis concerning the correlation ore "orthogonality" of these markers is provided, so for the reader it is difficult to understand the "true" value of the entropy markers. In summary, I would argue that the statistical analysis of the observed data is not fully convincing.
Reviewer 3 Report
Summary and general comments
This study aimed at investigating the application of RRI namely the Refined Composite Multiscale Entropy (RCME) analysis in healthy young people under cognitive tasks.
However, there are several questions need to be answered.
Major comments
The authors need to change the title. The output physiological signals are too wide. What is “new” in your study compared with others (model, design, idea, application, findings, conclusions)? Please state your hypothesis and aim(s) of the study clearly in the manuscript. Is the study designed (i.e. protocol) to test the hypothesis or fulfill your aims? Is your study designed (i.e. protocol) to test the hypothesis or fulfill your aims? Do the conclusions that you reach completely answer your original questions? Why Electroencephalography (i.e., EEG) is not used for this study for cognitive tasks?
Minor comments
The representation of the experimental procedure was not clearly presented? The authors need to show some papers in reference to the test. The ”old” markers (time and frequency domains) were addressed for comparison. The authors need to find some new methods for the study. The abstract should be more precise.
Round 2
Reviewer 2 Report
The new version of the paper is well structured, clearly written and easy to read. Everything seems to be technically sound. Most essential concerns in my first report have been addressed in a convincing way. In particular, the statistical evaluation of the results is much clearer now. I still do not think that this work will be a game-changer in this application domain, but it certainly provides a solid contribution to the problem of analyzing cognitive tasks with entropic measures.
Author Response
We thank the reviewer for the helpful comments.
Following the editor request, additional minor corrections have been performed.
Sincerely yours